# Pre-Diagnosis Health Seeking Behaviors and Experiences Post-Diagnosis, among Men Diagnosed with Tuberculosis in a District of Gauteng Metropolitan City, South Africa: In-Depth Interviews

**DOI:** 10.3390/ijerph192013635

**Published:** 2022-10-20

**Authors:** Sewele Makgopa, Lindiwe P. Cele, Mathildah M. Mokgatle

**Affiliations:** Department of Public Health, Sefako Makgatho Health Sciences University, Molotlegi Street, Pretoria 0208, South Africa

**Keywords:** male healthcare-seeking behavior, TB diagnosis, TB stigma, disclosure

## Abstract

Background: Tuberculosis remains the number one killer among infectious diseases in South Africa. The TB disease burden is said to be higher among males, 1.6 times more than females in 2018. Moreover, men are reported to have poor healthcare-seeking behaviors. Loss in social and physical functioning, including reduced sexual desires and changes in family life, have been reported following a TB diagnosis. This study explored the meaning that male TB patients attach to their TB diagnosis and impact of TB infection in their lives and those of the people living with them. Methods: This exploratory qualitative study was conducted among 25 participants recruited among male patients seeking TB care from two clinics in informal settlements of the city of Johannesburg. In-depth interviews with open-ended questions were conducted using an audio recorder for the collection of data. Data analysis was conducted on the NVivo version 12 software following an inductive thematic approach. Results: The ages of the participants ranged between 18 and 61 years. Most were unemployed, and only a few were married or in steady relationships. From the two emerging themes, pre-TB diagnosis health-seeking behaviors and post TB-diagnosis experiences, several subthemes were identified. For the former theme, the subthemes include, seeking help from community-based healers and self-medicating, waiting for some period to see if the alternative medicine or treatment worked, taking time to visit a healthcare facility, triggers to seek healthcare, and symptoms reported on presentation to the healthcare facility. The post-TB diagnosis subthemes include making sense of the TB diagnosis, context of disclosing the TB status, fear of social exclusion and experiences of stigma, support received during illness, life changes after TB infection and diagnosis, and lessons learned from the TB experience and future healthcare-seeking behavior. Conclusion: Secrecy about the TB diagnosis indicates fear of social exclusion, and this could be due to the highly stigmatized nature of TB. Waiting to see if alternative medication worked, delayed the TB diagnosis, with consequent late initiation of the anti-TB treatment. The life changes experienced post-TB diagnosis affect the quality of life of the participants and their families. The study recommends that these issues be addressed as a priority.

## 1. Introduction

Tuberculosis (TB) continues to be a major health problem around the world, with 10 million people reported to develop TB every year, globally. In South Africa, TB remains the number one killer from infectious diseases, contributing 6% to the 44% total deaths from natural causes in 2020 [1,2,3]. The country is among the top 8 countries that contributed two-thirds—of 30 countries, representing 87%—of the incident TB caseload in 2019 [4]. An estimated 328,000 new cases of TB are reported to have occurred in 2020, an incidence rate of 554 per 100,000 population [5]. This is despite concerted effort and strategies by the South African government to reduce the burden of TB disease among its populations. The country has made effective TB treatment available to its affected populations and scaled up TB diagnostics by being among the first countries to adopt the WHO-endorsed GeneXpert. This machine yields TB test results including rifampicin resistant TB (RR-TB) within 2 h [6].

The drivers of the South African TB epidemic are, among others, low socioeconomic status and high rates of HIV infections [2]. Additionally, South Africa remains a dual economy with one of the highest inequality rates in the world. This is driven by the heritage of exclusion and a low rate of economic growth that does not create jobs. Factors inherent in socioeconomic status, socio-demographics and behavior still play an important role not only in the outset and spread of TB, but also in its prevention, treatment and cure [2,7,8]. The South African majority live in poverty and crowded spaces with an unemployment rate of 33.6%, which is highest among youths between the ages of 15 and 24 years (63%) [9,10]. Additionally, impoverished people are less likely to seek healthcare or finish treatment, further contributing to failure in TB control. These conditions are more common in the Gauteng province, which is the economic hub of South Africa and the location of this study [11].

For TB control efforts to succeed, it is important to understand the way of living with the disease from the infected person’s perspective. According to Dr. Eric Goemaere (HIV–TB coordinator for MSF), if patients and doctors could speak up about the challenges, major barriers to TB control can be overcome [12]. With the burden of TB reported to be higher among males in South Africa and elsewhere in the world, almost 1.6 times more than in females in South Africa [2,13], this study sought to explore the meanings that male TB patients attach to their TB diagnosis. Additionally, men are reported to have poor healthcare-seeking behaviors [14,15]. Understanding the meanings people attach to their conditions can help inform the development of tailormade TB control strategies

## 2. Materials and Methods

### 2.1. Study Design

We conducted an exploratory qualitative study, using in-depth interviews (IDIs) to gather data on experiences of a TB diagnosis among male TB patients, attending two healthcare facilities in a metropolitan area of Gauteng.

### 2.2. Study Setting and Study Population

The two selected clinics serve communities of two informal settlements in the Johannesburg Metropolitan Municipality in Gauteng. The two informal settlements have the same pattern in their demographic makeup and economic status. They represent poor socioeconomic conditions characterized by overcrowding from informal dwellings, most of which are shacks, made of corrugated iron. The areas lack basic services such as water and sanitation. Alternative water and sanitation are provided through communal taps and toilets. The area has poorly constructed infrastructure, with formal dwellings making up only 30% of residential structures, and a concentration of taverns; which are some of the conditions that are conducive to high TB transmission rates. The population is young, with an average age of 25 years, mobile and migrant, mostly people from Zimbabwe and Mozambique. Unemployment rates are high, between 40% and 50%, with many households without an income [16,17]. The study population was male TB patients who were receiving TB care from the two clinics. They were included if they were 18 years old and above and on TB treatment at the time of data collection (inclusion criteria).

### 2.3. Sample Size and Sampling Technique

We employed a purposive sampling technique and selected between 20 and 30 male TB patients who met the inclusion criteria.

### 2.4. Data Collection and Participant Recruitment

The data were collected by the researcher and research assistant, trained in qualitative interviewing. Data collection took eight months, starting from July 2016 and ending in March 2017. Participant recruitment was done on the day of the consultation. Those who were willing to participate were given information about the study, and a verbal agreement was reached to conduct the interviews during the patient’s next visit to the facility. On the day of the interview, the researcher started by re-explaining the purpose of the study, including the expected duration of the interview and use of the audio recorder. Participants were then asked to sign the informed consent form before proceeding with the interviews. For data collection, the researchers used the TB treatment registers to collect information regarding the clinic visits for the selected participants and conducted the face-to-face interview in a private room, using a semi-structured interview guide, Table 1. The guide was translated into the two languages used in the area; languages in which the researcher was conversant. A digital audio recorder was used to gather all the conversations. Broad questions were about the participant’s life before and after the TB diagnosis, healthcare-seeking behavior, effects of disclosure and the changes in their lives since the TB diagnosis.

The socio-demographics of the participants including age, weight, nationality, employment status, level of education, marital status and number of children fathered were collected at the end of the face-to-face interviews. Each interview lasted approximately 30 min, and data saturation was reached when 25 participants had been interviewed. After data collection, all the data from the audio recordings were transcribed verbatim. The transcripts were then translated into English, typed and saved as Microsoft Word documents, and these were prepared for data analysis before being read repeatedly and coded manually for identification of common themes.

### 2.5. Data Analysis

The sociodemographic data were analyzed descriptively and presented in frequency tables. For the qualitative data analysis, the transcripts were repeatedly read to get familiarized with the data. After thorough familiarization with the content, initial coding was manually performed from the first five transcripts, and a codebook was developed. All the transcripts were uploaded into the QRS NVivo version 12 software where the codes were applied. Quotation and excerpts were coded, and themes and sub-themes emerged during the analysis.

### 2.6. Ethics Research Approval 

The study sought ethical clearance from the Research Ethics Committee of Sefako Makgatho Health Sciences University (SMUREC/H/162/2016), and permission to conduct the study was obtained from the Gauteng Department of Health district office and the research committee of the City of Johannesburg municipality (ref: 2016-17-054). The clinical managers granted access to participants for the interviews. Participants were requested to sign a consent form before proceeding with the interviews. Voluntary participation and withdrawal from the interviews at any point was emphasized to all participants. Pseudonyms were used during data collection and presentation of the quotations and excerpts.

## 3. Results

### 3.1. Sociodemographic Characteristics of the Participants

The demographic data indicated that the ages of the 25 participants ranged from 18 to 61 years with most (8/18) between the ages of 32 and 41 years. Most were South African (15/22), Sotho speaking (4/12), belonging to the Christian religion (12/18) and living in a shack (15/19). Only 5/19 and 3/19, respectively, were married and in steady relationships. The socioeconomic data revealed that nearly half were unemployed (8/17), while 6/15 had a regular salary, with another 6/15 relying on handouts. Most weighed less than 70 kg (17/20) and only (4/12) had family support. The majority had between one (1) and four (4) children (11/15) with half (7/14) having between grades 8 and 11 level of education [Table 2].

### 3.2. Findings from the Interviews

Table 3 displays the two themes that emerged and the subthemes that were identified. From pre-diagnosis health-seeking behaviors, five subthemes were identified, and they are: seeking care from community-based healers and self-medicating, waiting for some period to see if alternative medicine or treatment works, taking time before presenting to the healthcare facility, triggers for seeking healthcare and symptoms reported on presentation to the healthcare facility. From the post-diagnosis behaviors and experiences, six subthemes were identified, and they are: making sense of the TB diagnosis, context of disclosing the TB status, fear of social exclusion and experiences of stigma, support received during illness, life changes after TB infection and diagnosis, and lessons learned from the TB experience and future healthcare seeking behavior.

#### 3.2.1. Pre-Diagnosis Healthcare-Seeking Behavior and Experiences

It was common practice among the men in this study, even though they felt sick, to first seek healthcare from community-based healers and take self-medication; only to visit the health facilities once the symptoms got worse, thereby delaying the formal diagnosis of TB. The following subthemes illustrate the delay in seeking healthcare:

##### Seeking Care from Community-Based Healers and Self-Medication

When participants got ill, they first sought help from non-medical providers; church, traditional healers and self-medicating with anything that they thought would bring relief from the illness:


*Most of the time when I feel like I have a problem, before I go to churches, I sometimes start by going to the traditional healers; and they would help me in whatever way they help me, and then I would feel alright. So, because they say they can treat those things, (HIV/TB) maybe they can. I am not sure.*
(55-year-old, James)

##### Waiting for Some Period to See If Alternative Medicine or Treatment Works


*When I felt sick I just bought some painkillers like Panado (paracetamol), and other medication, to alleviate a bit on the pains. …Yes, they would just assist a bit with what I was feeling. … they helped just a little bit, just a little bit but the sweating was still persisting.*
(37-year-old, Hope)

##### Taking Time before Presenting to the Healthcare Facility

It took between 3 weeks and 3 months for the participants to visit the healthcare facility to finally get the TB diagnosis:


*It took me a while because I didn’t know it was TB related …I can say about 3 months after my HIV diagnosis because I didn’t believe that it could be something related to TB. I stayed at home all that time sick …until I decided that let me go and check for TB as well.*
(43-year-old, Goodwell)


*When I put it accurately, it could be 1 month and 2 weeks when we combine all that period from the doctor, at work when I was beginning to feel ill…and all that.*
(45-year-old, Peter)


*No, it was 3 weeks, it was roughly 3 weeks.*
(40-year-old, Gilbert)

##### Triggers for Seeking Healthcare

While a few were motivated by close acquaintances, most only visited the healthcare facility when the symptoms became worse:


*I think close to 7 people came up to me and ask if I don’t have money to go to the hospital so that they give it to me. Which means they were telling me that I should go and get tested.*
(43-year-old, Goodwell)


*I was getting weak, had difficulty in breathing, not sleeping, sweating, coughing you see.*
(31-year-old, unemployed, Jabu)


*I kept on going to church because I believed that those things (the symptoms) will get better as I am busy going to church. As time went on I noticed that, no, these things are becoming worse a bit, and I thought that the best thing is that I should come to the clinic for a thorough checkup.*
(57-year-old, Nkosi)

##### Symptoms Reported on Presentation to the Healthcare Facility

The symptoms that the participants experienced varied from one person to another. While others experienced the classical TB symptoms, for some the symptoms were non-specific for TB:


*I just saw it as flu; as the days progressed I was getting weaker, wasn’t sleeping, and I was sweating.*
(18-year-old, matriculant, Albert)


*I was getting weak, had difficulty in breathing, not sleeping, sweating, coughing you see.*
(42-year-old, Shaun)

#### 3.2.2. Post-Diagnosis Behaviors and Experiences

Once the TB diagnosis was confirmed, there was a phase in which the men reported that they had to make sense and understanding of their disease condition, had to consider how to disclose the TB status, feared judgement and experienced stigma from many angles, and had to consider disclosing their TB diagnosis to significant others and receiving support. The men in this study further reported that they experienced some changes in their lives because they were weaker than before. They mentioned that they have learned to avoid delays in reporting their health conditions to healthcare providers.

##### Making Sense of the TB Diagnosis

When participants became aware of their TB status, some expressed shock and disbelief and others started questioning themselves about where they got it. While others thought smoking and witchcraft were the sources of their illness, some started connecting the TB to HIV infection.


*I asked myself where I could have gotten it. You see that question I did ask myself…where this TB is from as I only came to the hospital because I was losing weight. My body was wasting. You see this sickness… there are people who think that I am not sick because I don’t feel any pain.*
(45-year-old, Mthembu)


*Eish, you see that made me think a lot. I thought it was a lot of things. I thought of witchcraft, maybe it is stress, or maybe someone has cursed me you see. But there was a part of me that also thought that it could be TB because of the relationship between TB and HIV.*
(31-year-old, unemployed, Jabu)

##### Context of Disclosing the TB Status

The participants expressed mixed feelings about disclosure of the TB status. While some developed the courage to disclose, with others disclosing to protect and benefit other people, others were outright opposed to disclosing:


*I built the courage and one day decided to sit her down and tell her that I am sick. So, I sat my woman down and told her what is happening in my life, and she accepted. There is one thing she emphasized though was the importance to take my treatment.*
(31-year-old, unemployed, Jabu)


*No, I do not tell anyone anything, no-no. I cannot tell another person my secret. The secret is mine with my wife and with my children, and that is finish and quiet. Even my friends I do not tell them.*
(57-year-old, Nkosi)


*It is important to inform people that you can have TB even if you are not coughing, and not doing things like that (not having symptoms); you can infect other people.*
(30-year-old, Marcus)

##### Fear of Social Exclusion and Experiences of Stigma

Most participants expressed occasional feelings of discrimination or exclusion from the community, family members, friends and their significant others including people at the healthcare facilities. They felt like they were judged and perceived in relation to the TB diagnosis:


*They hit me with words, others were even talking about AIDS and what what…you see stuff like that…”.*
(55-year-old, Peter, unemployed)


*But then like, when you come to the clinic, a lot of people looks at you that you have come…what are you here for? [Clears throat] because they know that which room is for people who take ARVs. And, [clears throat] that thing makes me scared a bit. [Clears throat] it makes me think that people will talk about me in a bad way and eventually everybody will know why I was here.*
(18-year-old, Albert, co-infected, matriculant)

##### Support Received during the Illness

Most participants indicated having some kind of support from people close to them, while a few others did not have that privilege of support:


*The kind of support that I am getting is the one from my friends. They occasionally take me out and we go and try to organize something to eat and then come back.*
(30-year-old, Marcus)


*I need support in the form of food and cosmetics, those are the main things. I struggle a lot because of them and I think about them constantly.*
(48-year-old, Sfiso)

##### Life Changes after TB Infection and Diagnosis

Participants experienced change due to physical effects of weight loss, tiredness and pain. Participants reported sexual and lifestyle changes including daily activities such as socializing, drinking alcohol, smoking, working, and participating in other social events:


*I was actually drinking with timing (occasionally), maybe I could drink a bit of two or three and then feel that, haai, I am fine because I took a long time not drinking alcohol. Alcohol is no longer there in my body and I am no longer used to having those things in my body. My body is used to just sitting like this.*
(37-year-old, Mogale)


*Well, there were changes even then because there would be times when I would try to sleep with her but I would be very easily fatigued to even attempt anything. I’m usually not used to being tired that quick, so this would really concern me.*
(31-year-old, Jabu)

##### Lessons Learned from the TB Experience and Future Healthcare-Seeking Behavior

Data show a consensus among all the men as they stated that the primary healthcare facility would be the first place to start seeking help, should they be infected with TB or be sick again in the future:


*Clinic. Clinic. I prefer clinic because they will tell me the truth. They will only tell you something that is really happening; they won’t hide anything. No pastors, no fortune-tellers my sister; those ones may come after. For me, it will be just the clinic.*
(52-year-old, Sabata)


*In future, if I can feel the same problem? What I have felt before…yah, I am going to the clinic. Yes, it is not that the church…the church is also able to work hand in glove with the clinic. Yes, it doesn’t mean the church…the church is also able to…it doesn’t mean that it is only about laying hands on you. They don’t give you the treatment. They don’t affect treatment, they…you see honestly the church and the clinic can work together. What I am not sure about is the traditional doctors because those ones will give you medication. Maybe you will find that they disturb each other; they (traditional treatment) conflict with the pills. So, but with the church and the clinic, I don’t see them having a problem.*
(43-year-old, Mulalo)

## 4. Discussion

This study explored the meaning of a TB diagnosis among male TB patients and found that a TB diagnosis caused shock and disbelief among the participants. Getting a positive diagnosis of a dreaded disease such as TB can create denial and cause an array of psychological reactions as patients worry about the effects of disclosing, fear of seclusion, suicidal thoughts and anxiety [18,19]. Participants in this study associated the illness with smoking and witchcraft, among others, while others started thinking of how TB is a proxy for HIV infection. A similar reaction was observed in a study that involved migrant African males in London, where participants indicated how the TB diagnosis precipitated HIV testing [20]. Participants in this study self-treated with painkillers, saw private doctors and sought help from church and traditional healers when they experienced the symptoms. Other studies have also reported participants seeking care first from non-medical providers before visiting the healthcare facilities [21,22,23]. The time taken to visit a health facility, which ranges from 3 weeks to 3 months, far exceeded the period that is stipulated for initiating TB treatment following the symptom onset [6]. Other studies have reported much higher time delays of up to 9 years [22,24,25]. This long delay is due to participants seeking help, first from non-medical providers, self-medicating and attributing the illness to other causes but TB. Linking TB symptoms to other causes has been cited as a barrier to the diagnosis and treatment of MDR [20,26]. However, other participants in the present study experienced flu-like symptoms, feeling weak, pain in the leg and headache, which are not the classic symptoms for TB. Similarly, participants in other studies did not recognize the TB symptoms and therefore delayed in seeking TB care [20,21,23,26].

While a few participants were motivated by other people to seek care from a health facility, others only visited a healthcare facility once they realized that the symptoms were not subsiding. Similarly, other studies have found a delay in seeking TB care because the participants did not feel ill, even though they were presenting with the TB symptoms [23]. The majority of participants preferred not to disclose to people outside of their family, expressing fear of discrimination. Similarly, other studies have reported non-disclosure due to fear of a backlash from colleagues who themselves did not want to contract TB from them. Non-disclosure can be due to the stigma that is attached to TB. In a national survey on stigma, about 34% of respondents admitted to not sharing their TB disease status to people outside of their homes, with 36% reporting being teased and sworn at because of their TB status. Others have reported isolation by work colleagues, and this is possibly due to the infectiousness and contagiousness of the TB disease [12,27]. On the contrary, some of the participants from the current study reported receiving support from family members and work colleagues. The support ranged from social visits by friends and work colleagues to finance and encouragement by family members to take the medication. Other studies have also found high support from family and friends among male participants [28].

The narratives on the life before and after revealed that the TB diagnosis and treatment brought some changes in the lives of the participants. While some reported stopping smoking, drinking and taking walks, and resorting to staying at home and doing house chores, others reported not engaging in sexual activities since the TB diagnosis and TB treatment; reporting exhaustion as the reason for this change. Similarly, other studies have found loss in social and physical functioning including reduced sexual desires and changes in family life with a TB diagnosis [29,30].

## 5. Limitations

The small sample size and non-random sample used in this study limit the generalizability of the findings to the target source population.

## 6. Conclusions

The narratives from this study indicate that a positive TB diagnosis had a negative effect in the lives of the male TB patients. Shock, disbelief, feelings of discrimination, experiences of finger pointing and gossip, including the non-disclosure are all effects of stigma. The TB stigma has been shown to increase delay in TB diagnosis and treatment non-compliance [31,32]. Moreover, TB stigma is thought to cause a sense of guilt or shame among TB patients, leading them to self-isolate as they internalize the negative judgements made by the community about the disease. Additionally, the social stigma-related psychological issues worsen the quality of life among patients with TB [20,32,33,34,35]. This study reiterates the urgent need for individual, family and community education strategies to be developed, especially in areas of high TB endemicity. This will help reduce the stigma attached to TB.

The social withdrawal and reduced sexual desires can have a negative impact in the participant’s quality of life, and therefore should be addressed. TB, both pulmonary (PTB) and genital, is said to disrupt the sexual function in both males and females; PTB has been associated with a decline in all parameters of the copulatory act; however, the number of anti-TB drugs taken simultaneously is thought to influence this process and not the disease itself. Male sexual function is vital for the sustenance of a marriage; the lack thereof has been reported to cause relationship breakdown [36,37].

The support received from family and close acquaintances requires encouragement and strengthening through targeted community awareness. Findings from other studies have indicated the importance of this kind of support in the treatment process as it contributes to treatment adherence and discontinuation, due to a lack thereof [38,39]. Visiting a healthcare facility only when the symptoms worsen requires urgent attention as this prolongs the time delay in seeking TB care. A delay in seeking TB care has been associated with increased risk of unsuccessful outcomes, prolonged patient suffering and providing the opportunity for increased transmission to contacts [39,40,41]. Consequently, late presentation to a healthcare facility undermines the requisite of early diagnosis and early initiation in effective TB therapy for the effective control of TB [6,42].

## Figures and Tables

**Table 1 ijerph-19-13635-t001:** In-depth interview questions.

**1. When you got sick, what did you think was the problem?**
What did you think was the cause of your illness?
**2. What was your life like before you got sick?**
**3. What did you do to get help with your sickness at that point (before going to the health facility)?**
What kind of help did you receive? Share the details.
How did that help you with the sickness?
**4. What made you decide to come to the clinic?**
**5. Were you tested for HIV prior or post your TB diagnosis? (If yes, when [prior or post?], and what were the results?)**
What did you think of the results?
**6. What do you think of having both HIV and TB at the same time?**
How do you think you can prevent the spread of both?
Which symptoms of TB/HIV are you aware of?
Which modes of spread of TB/HIV are you familiar with?
**7. Please share with me what went through your mind the minute you heard that you had TB.**
**8. Please share with me how you are feeling now since your diagnosis.**
**9. What do you think of the possibility of infecting your contacts with TB?**
If HIV positive, what do you think of the possibility of infecting your sexual partner/s?
Are there any measures you take to avoid infecting your family/partners/contacts with TB/HIV (please elaborate)?
**10. How did the clinic assist in tracing your contacts/partner(s) for screening of TB/HIV?**
**11. What has changed in your life since you have known that you have TB/HIV? How do the changes make you feel?**
**12. What has changed in your life since you have been on treatment?**
Effect of treatment
Work relationships
Community reaction
Coping with work
Intimate relationships: How do you currently engage with your partner sexually (fear of infection, libido, physical fitness)?
Being co-infected
**13. Are you familiar with other ways of treating TB/HIV other than with medical treatment? Please share.**
If there are other ways you are familiar with, which one do you think is the best? Elaborate.
What do you think of taking dual treatment (HIV/TB treatment)?
What affects your adherence to your treatment?
How do you think disclosing affects adhering to treatment?
**14. In your community, how is TB/HIV perceived?**
How is the perception different from the one you had before your diagnosis?
How does it feel to be on the receiving end of the community’s perception?
What is the perception of your family/partner/contacts?
**15. What kind of support do you need now since the diagnosis? What kind of support are you getting and from whom?**
**16. If you could change some things about TB/HIV care and treatment, what would they be?**
**17. In your own words, what would you tell a new TB/HIV patient about this condition/s and the treatment?**
How would you encourage them to comply with their treatment?
**18. Considering the total amount of time you took before coming to the clinic, in the future, will you change your health-seeking behavior? Elaborate.**
**19. In the future if you get infected with TB again, where would you go first to seek help?**

**Table 2 ijerph-19-13635-t002:** Sociodemographic characteristics.

Variable	Sub-Category	Frequency
Age group, *n* = 18		
	18–21	1
	22–31	3
	32–41	8
	42–51	4
	52–61	1
Nationality, *n* = 22		
	South African	15
	Zimbabwe	3
	Lesotho	4
Ethnic group, *n* = 12		
	Venda	2
	Sotho	4
	Sepedi	3
	Ndebele	3
Relationship status, *n* = 19		
	Married	5
	Single	3
	Casual relationship	7
	Steady relationship	3
	Cohabiting	1
Belief system, *n* = 18		
	Christian	12
	Traditional	4
	None	2
Type of dwelling, *n* = 19		
	Shack	15
	RDP	2
	Back room	2
Employment status, *n* = 17		
	Formal employment	6
	Informal employment	2
	Unemployed	8
	Self-employed	1
Source of support, *n* = 12		
	Self	2
	Wife/partner	3
	Family	4
	Friends	2
	None	1
Occupants in dwelling, *n* = 17		
	1	1
	2	7
	≥3	5
	None	4
Income, *n* = 15		
	Regular salary	6
	Irregular salary	2
	Handouts	6
	Grant	1
Children, *n* = 15		
	1–4	11
	≥5	2
	None	2
Level of education, *n* = 14		
	No education	0
	Primary school	1
	Grade 8–11	7
	Matric	6
Weight, *n* = 20		
	40–49 kg	6
	50–59 kg	5
	60–69 kg	6
	70–79 kg	3

**Table 3 ijerph-19-13635-t003:** Themes and subthemes.

**3.2.1 Pre-Diagnosis Health-Seeking Behaviors**	**3.2.2 Post-Diagnosis Behaviors and Experiences**
**3.2.1.1 Seeking care from community-based healers and self-medication**	**3.2.2.1 Making sense of the TB diagnosis**
Bought pain killers such as Panado and took traditional medicines Attended church prayers and visited traditional healers	Wondered about how one got infected Tried to figure out what caused the infection
**3.2.1.2 Waiting for some period to see if alternative medicine or treatment works**	**3.2.2.2 Context of disclosing the TB status**
Got temporal pain relief from self-medication	Developed the courage to disclose Resorted to secrecy and selective disclosure to significant others Decided to disclose in order to protect and benefit others
**3.2.1.3. Taking time before presenting to the healthcare facility**	**3.2.2.3 Fear of social exclusion and experiences of stigma**
Took between 3 weeks and 3 months to visit the healthcare facility following symptoms	Experienced name calling related to the ill-health Feared being judged and devalued based on clinic visits
**3.2.1.4 Triggers for seeking healthcare**	**3.2.2.4 Support received during illness**
Got advice from acquaintances Symptoms got worse	Social visits by friends and work colleagues Finance and encouragement by family members, to take the medication.
**3.2.1.5 Symptoms reported on presentation to the healthcare facility**	**3.2.2.5 Life changes after TB infection and diagnosis**
Flu-like symptoms such as difficulty in breathing and feeling weak Classic clinical features of TB infection	Stopped smoking and alcohol consumption Experienced reduced sexual desire and sexual performance Stopped taking walks, and resorted to staying at home and doing house chores
	**3.2.2.6 Lessons learned from the TB experience and future healthcare-seeking behavior**
	Taking care of oneself Visiting a healthcare facility first, when feeling ill to get the truth

## Data Availability

The data files generated and analyzed during the current study are available from the corresponding author upon request.

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
