# Peer review of "Pre-Diagnosis Health Seeking Behaviors and Experiences Post-Diagnosis, among Men Diagnosed with Tuberculosis in a District of Gauteng Metropolitan City, South Africa: In-Depth Interviews"

_ijerph, 2022, doi:10.3390/ijerph192013635_

Round 1

Reviewer 1 Report

Dear authors!

This contribution is valuable and impactful, especially because of its population. Congratulations for having reached such a "hidden" segment of population.

The paper is almost ready. There are two suggestions I would like to make:

1. In the introductory part, TBC was linked to HIV-infection, 58% of cases were said to be associated. However, from the interviews this link remains unclear. There are some hints, also some interview excerpts that SUGGEST this, but the authors have not extract this point from the interviews to elucidate it more, it remains in the shadow. Please make this point much clearer in the Discussion section, because it will make the emotions of fear and stigmatization much more understandable to the reader.

2. Overpunctuation is present in the paper, two many commas are used on inappropriate paces. Please make a sorrough stylistic proofreading.

Good luck with the continuation.

Best wishes,

Reviewer

Author Response

Dear reviewer

Thank you for reviewing our our manuscript. Kindly see the attachment for the responses 

Regards

Reviewer 2 Report

The manuscript under review is an article on the impact of TB diagnosis on patients and family members’ life. It has been conducted through in-depth interviews to 25 participants. The paper is well-written. However, I have a few comments:

-          The introduction is quite long, I suggest the Authors reducing it a little bit, eliminating unnecessary information.

-          If possible, I suggest adding a figure or a table representing the questions asked during the in-depth interviews.

-          I suggest implementing the references list with the following: 10.1164/ajrccm.164.1.2101091; 10.3390/healthcare10020386; 10.7860/JCDR/2016/19678.8598

Author Response

Dear reviewer

Thank you for reviewing our manuscript. Kindly see the attached for the responses

Regards

corresponding author
